# Comparable Accuracy of Quantitative and Visual Analyses of [^18^F]FDG PET/CT for the Detection of Lymph Node Metastases from Head and Neck Squamous Cell Carcinoma

**DOI:** 10.3390/diagnostics13162638

**Published:** 2023-08-10

**Authors:** Philippe d’Abadie, Nicolas Michoux, Thierry Duprez, Sandra Schmitz, Michèle Magremanne, Pascal Van Eeckhout, Olivier Gheysens

**Affiliations:** 1Department of Nuclear Medicine, Cliniques Universitaires Saint-Luc, Institut Roi Albert II, Université Catholique de Louvain, 1200 Brussels, Belgium; olivier.gheysens@uclouvain.be; 2Department of Radiology, Cliniques Universitaires Saint-Luc, Institut Roi Albert II, Université Catholique de Louvain, 1200 Brussels, Belgium; nicolas.michoux@uclouvain.be (N.M.); thierry.duprez@uclouvain.be (T.D.); 3Department of Head and Neck Surgery, Cliniques Universitaires Saint-Luc, Institut Roi Albert II, Université Catholique de Louvain, 1200 Brussels, Belgium; sandra.schmitz@uclouvain.be; 4Department of Stomatology and Maxillofacial Surgery, Cliniques Universitaires Saint-Luc, Institut Roi Albert II, Université Catholique de Louvain, 1200 Brussels, Belgium; michele.magremanne@uclouvain.be; 5Department of Pathology, Cliniques Universitaires Saint-Luc, Institut Roi Albert II, Université Catholique de Louvain, 1200 Brussels, Belgium; pascal.vaneeckhout@uclouvain.be

**Keywords:** FDG PET/CT, head and neck squamous cell carcinoma, nodal staging, TNM classification, diagnostic accuracy

## Abstract

Background: In head and neck squamous cell carcinoma (HNSCC), [^18^F]FDG PET/CT is recommended for detecting recurrent disease and in the initial staging for evaluating distant metastases, but its use in detecting cervical lymph metastases remains unclear. The aim of this study is to evaluate and compare the diagnostic accuracy of [^8^F]FDG-PET/CT using visual and semi-quantitative analyses for detecting the nodal involvement in HNSCC. Methods: We analyzed consecutive patients who underwent a preoperative [^18^F]FDG-PET/CT and neck dissection for HNSCC at our tertiary hospital. A blinded evaluation of the [^18^F]FDG uptake in each neck level was performed using a semi-quantitative approach (SUVmax and SUVR) and a visual grading system (uptake superior to the internal jugular vein for grade 1 and superior to the liver for grade 2). Analyses were compared to the histological results. Results: In our 211 patients, analyses demonstrated similar diagnostic accuracy using a semi-quantitative approach or a visual grading system. Regarding the visual grading system, [^18^F]FDG-PET/CT detected nodal metastases with a specificity of 83% for lymph nodes classified as grade 1 and 98% for those classified as grade 2. The sensitivity was moderate, ranging from 60 to 63%. Conclusions: [^18^F]FDG PET/CT has a high specificity for detecting lymph node metastases in HNSCC and therefore must be considered in the nodal clinical staging.

## 1. Introduction

Head and neck squamous cell carcinoma (HNSCC) is the sixth most common cancer worldwide and constitutes 5–10% of all new cancer cases in Northern America and Europe [1,2]. The most common sites are the oral cavity (≈44%), the larynx (≈31%) and the oropharynx (≈25%) [3]. The prognosis of HNSCC has significantly improved in recent decades, resulting from a significant improvement of diagnosis, staging and treatments [4,5]. The initial staging based on the Union for International Cancer Control (UICC) and the American Joint Committee on cancer (AJCC) Tumour Node Metastases (TNM, 8th edition) determines the treatment strategy and prognosis [6,7].

For this purpose, imaging plays a central role for accurately evaluating the extent of the primary tumor, the presence of cervical lymph node involvement and distant metastases. Conventional imaging, including magnetic resonance imaging (MRI) and computed tomography (CT), is recommended for evaluating the local extension and regional lymph node status. Approximately 60% of HNSCC are locally advanced at time of diagnosis (T3-T4) and are associated with a higher risk of metastases in regional lymph nodes and at distant sites [8]. An accurate evaluation of lymph node involvement is crucial before treatment because the 5-year survival rate decreases to less than 50% for patients with lymph node metastases [9]. The number of positive lymph nodes is also a strong prognostic factor in HNSCC patients [10]. The recent literature reports a variable diagnostic accuracy of conventional imaging, with a sensitivity and specificity ranging from 77% to 100% using enhanced MRI [11]. Despite significant improvements in conventional imaging, the overall rate of occult micrometastases, undetectable by conventional imaging, ranges between 10% and 55% [12].

Consequently, selective neck dissection of the possible involved lymph node regions is generally performed based on the predicted dissemination pathways of the tumor site [13]. Oral cavity tumors are mainly drained in ipsilateral neck levels I to III, while oropharyngeal and laryngeal tumors are drained in ipsilateral levels II to IV. A bilateral neck dissection is often considered for tumors with a risk of spreading to the contralateral neck (close to or midline tumor, base of the tongue or supraglottic laryngeal tumors). Selective neck dissection maximizes disease-free survival but is associated with a significant risk of complications and possible impaired quality of life [14,15]. Therefore, non-invasive screening methods that reliably predict N0 status may reduce this rate of futile interventions. [^18^F]FDG PET/CT has proven useful for the evaluation of distant metastases in HNSCC, but its accuracy for the detection of lymph node metastases remains unclear [16,17,18]. Moreover, standardized PET interpretation criteria to define this nodal status are currently lacking and this might help to improve the accuracy of the technique.

The aim of the present study was therefore to evaluate and compare the diagnostic accuracy of [^8^F]FDG PET/CT for identifying lymph node metastases in HNSCC using visual and semi-quantitative methods.

## 2. Materials and Methods

### 2.1. Population

We retrospectively analyzed data of all consecutive patients who underwent a preoperative [^18^F]FDG PET/CT within a 30-day interval before neck dissection for histologically proven HNSCC de novo or recurrence at our institution between 2007 and 2019. Patients with chemotherapy induction or previous treatment with radio-chemotherapy within 6 months before surgery were excluded from the analysis. Our multidisciplinary tumor board classified the clinical neck stage (cN staging) for each patient before surgery, based on physical examination and conventional imaging (cN0 or cN+). Gadolinium-enhanced MRI of the neck was performed for each patient. Patients classified as cN+ had at least one lymph node with a short axis higher than 10 mm. No other criterion was used for determining the cN status. Our local ethics committee (2019/02AOU/347) approved the study.

### 2.2. [^18^F]FDG PET/CT Acquisition and Analysis

Whole-body PET/CT imaging was performed on a Gemini TF (Philips Medical Systems, Cleveland, OH, USA). Before imaging, patients fasted for at least 4 h and blood glucose levels were confirmed <200 mg/dL before intravenous administration of 280–310 MBq [^18^F]FDG. PET emission was acquired 60 min later, preceded by a low-dose CT with a 120 kV dose modulation protocol from the vertex to mid-thigh. All acquisitions were anonymized and exported in a computer server.

A certified nuclear medicine physician, blinded to all clinical data, recorded the presence and [^18^F]FDG uptake in lymph nodes in neck levels I to V for each patient. First, a visual analysis was performed using a 3-point grading scale, comparing the lymph node uptake to those of the internal jugular vein and of the liver. A lymph node classified as grade 0 had no significant uptake (≤blood pool activity in internal jugular vein), a lymph node classified as grade 1 had a low uptake (>blood pool activity in internal jugular vein and ≤liver uptake) and a lymph node classified as grade 2 had a high uptake (>liver uptake). Figure 1 illustrates examples of lymph nodes classified as grade 1 (Figure 1A) and grade 2 (Figure 1B). Secondly, [^18^F]FDG uptake in lymph nodes was quantified using the maximum standardized uptake value (SUV max) and a lymph-node-to-liver ratio (SUV max _lymph node_/SUV mean _liver_) named the SUV ratio (SUVR).

### 2.3. Surgery and Reference Standard

The surgical procedure was performed in line with the institutional standards and guidelines. For each patient, a detailed report of the neck dissection was made, including the operated neck levels, the precise localization of the metastatic lymph nodes (neck level), the number of resected lymph nodes and, if available, the size of the lymph node metastases.

The neck levels with histologically proven lymph node metastases were considered positive. The neck levels with negative histology or those not considered for neck dissection by the surgeon were considered negative. The [^18^F]FDG uptake was correlated to the size of the lymph node metastases (when available in the histological report).

### 2.4. Statistical Analysis

Performance of the visual grade (0/1/2), SUV max and SUVR in each neck level was assessed by their respective Receiver-Operating Characteristic (ROC) curves, using the histology report as the gold reference. Sensitivity (Se), specificity (Sp), cut-off criterion and Area Under the ROC Curve (AUC) were reported with their 95% Confidence Interval (95% CI) and *p*-value of the test. Then, for a given group (cN0, cN+ and the complete cohort), the performance of the parameters was compared against each other using the DeLong test for comparing AUCs [19]. An Exact 2-sided test was performed to compare the Se and Sp of each parameter (SUV max, SUV ratio, visual grade) according to the cut-off criterion that was chosen (comparison of paired proportions), as well as in groups cN0 versus cN+ (comparison of independent proportions) [20].

As the distribution of lymph node sizes was non-normal (according to the Shapiro–Wilk test), a Mann–Whitney U-test was used to compare the median lesion size according to the visual grade. The correlation between the lesion size and [^18^F]FDG uptake was also investigated using the Spearman rank correlation coefficient.

A *p*-value < 0.05 was regarded as statistically significant for all tests cited above. The statistical software Medcalc V20.014 (https://www.medcalc.org/ (accessed on 20 May 2023)) was used for the analysis.

## 3. Results

### 3.1. Patient Characteristics

A total of 211 patients (72% men, median age 61 years) were included in the study. The main patient characteristics are reported in Table 1. In summary, the majority of patients were treated for oral cavity HNSCC (156 patients, 74%) and laryngeal HNSCC (40 patients, 19%). Among the 211 patients, 145 patients (69%) were classified as cN0 and 66 (31%) as cN+.

Regarding the histological results, 122 patients (58%) had early-stage T1-T2 tumors, of whom 43 patients (35%) had lymph node metastases. Eighty-nine patients (42%) had locally advanced tumors (T3-T4), with lymph node metastases in forty-six patients (52%). The main results of the neck dissections are summarized in Table 2.

### 3.2. Diagnostic Performance

The diagnostic performance of SUV max, SUVR and the visual scale for detecting lymph node metastases (per neck level) is displayed in Figure 2 and Table 3.

For the whole cohort of 211 patients, visual and semi-quantitative indices yielded similar sensitivity (60–63%) and specificity (86–88%), with a cut-off SUV max > 1.9, SUV ratio > 1.06 and visual grade > 0, respectively. Using higher cut-off values (SUV max > 3.6, SUV ratio > 2.1, grade > 1), specificity increased significantly (SUV max: +13% [+11%; +14%]; SUVR: +11% [+10%; +13%]; grade: +12% [+11%; +14%], but at the cost of a significantly attenuated sensitivity (SUV max: −22% [−15%; −29%]; SUVR: −29% [−21%; −36%]; grade: mean difference = −19% [−13%; −26%], *p* < 0.0001 for all). Figure 3 shows an example of a patient with a lymph node metastasis corresponding to a [^18^F]FDG uptake grade 1.

Regarding the whole cohort, the SUVmax of lymph nodes with proven metastases at histology were significantly higher than the SUVmax of lymph nodes without proven metastases (Table 4). Note that the SUVmax of malignant lymph nodes were generally higher than the SUVmax of the liver, and the SUVmax of benign lymph nodes generally lower than the SUVmax of the liver (comparing the means).

Subgroup analysis comparing patients with cN0 to cN+ showed significantly lower sensitivity for detecting nodal metastases in patients with cN0 compared to cN+, regardless of the parameter used (Table 5).

### 3.3. Correlation between Morphology and PET Metrics

In 20 of 211 patients, the size of lymph node metastases (*n* = 69) was available for analysis. A significantly positive correlation was observed between the size and the [^18^F]FDG uptake (SUV max) in the corresponding neck level (rho = 0.50 [0.30; 0.66], *p* < 0.001). As shown in Figure 4, the median size of metastases was significantly higher for lymph nodes with SUVmax > 1.9 (10 mm) compared to lymph nodes with SUV max ≤ 1.9 (3 mm, *p* < 0.001).

As shown in Figure 5, the median sizes of metastases classified as visual grade 0, 1 or 2 were, respectively, 3.00 mm [1.17 mm; 5.00 mm], 8 mm [5.03 mm; 11.9 mm] and 12.0 mm [11.2 mm; 21.2 mm]. The median size in visual grade 0 was significantly lower compared to visual grades 1 and 2 (*p* = 0.0298 and, *p* = 0.0002, respectively). For information, one discordant finding was demonstrated in Figure 5, with a lymph node metastasis of 18 mm with an unusual grade 0 uptake (shown as case 15). For this patient, the PET reader classified a grade 2 uptake in the neck level II and the histological report classified this large metastasis in the level III.

## 4. Discussion

In a large cohort, this study demonstrated the high specificity of [^18^F]FDG PET/CT for detecting lymph node metastases with a specificity of 86% using a visual score grade 1 and a specificity of 98% for a grade 2. These results are concordant with previous studies reporting specificities ranging from 86% to 96% [16,18]. As such, our study shows that every lymph node with an uptake higher than vascular blood pool in a predicted pathway neck level of the primary tumor must be considered as suspect and that a visual grade 2 is strongly correlated with malignancy. In our data, the false positive rate was only 1–2% for lymph nodes with high [^18^F]FDG uptake (SUVmax > 3.6 or visual score grade 2). For lymph nodes with low [^18^F]FDG uptake (SUVmax > 1.9 or visual score grade 1), the false positive rate increased to 14%. Therefore, the higher the [18F]FDG uptake in the lymph nodes, the more likely the malignancy. In lymph nodes with low [^18^F]FDG uptake, false positive results due to reactive lymph nodes cannot be excluded. Our results are concordant with the study of Nakagawa et al., demonstrating a low [^18^F]FDG uptake in reactive lymph nodes (mean SUVmax = 2.2) [21].

Regarding the sensitivity, [^18^F]FDG PET/CT demonstrated a moderate accuracy ranging from 60 to 63%. These results are in line with earlier findings on the performance of [^18^F]FDG PET/CT for detecting lymph node involvement. A meta-analysis by Sun et al. reported an overall pooled sensitivity of 80% using a similar neck level analysis, but a large heterogeneity in sensitivity was reported for individual studies with values ranging between 31% and 100% [16]. A prospective study including 91 patients with similar characteristics, mostly oral cavity cancer and nodal stage pN0, reported a sensitivity of 69%, similar to the one reported in this study [22]. Regarding subgroup analysis comparing the preoperative cN status, we showed a significantly lower sensitivity of [^18^F]FDG PET/CT in cN0 patients compared to cN+ (40% vs. 75%). This study confirms earlier findings on the lower sensitivity for detecting lymph node involvement in cN0 patients. Indeed, a meta-analysis performed by Kyzas et al. reported a pooled sensitivity of 50% in cN0 patients versus 94% for cN+ [18]. This can be attributed to the fact that patients classified as cN+ have larger metastatic lymph nodes, often palpable and larger than 10 mm compared to patients classified as cN0. The performance of [^18^F]FDG PET/CT is limited by its inherent spatial resolution. Phantom studies performed on a similar Philips Gemini PET/CT system have shown that the recovery coefficient dropped exponentially with smaller sphere diameter, and even stronger for those with a diameter below 10 mm [23]. The latter explains that [^18^F]FDG uptake is strongly underestimated in small metastatic lymph nodes, and hence small lymph node metastases are not detected on PET. Therefore, small or micro metastases can have a low or no significant uptake compared to the jugular internal vein, leading to a visual score of grade 0. This limitation is supported by Figure 4, which illustrates a visual grade 1 uptake in in lymph nodes with a median size of 8 mm and an absent uptake (visual grade 0) for those with a median size of 3 mm. Nevertheless, our data demonstrated that [^18^F]FDG PET/CT detected lymph node metastases in up to 40% of neck levels classified as negative using conventional imaging (i.e., lymph nodes with a short axis < 10mm), and therefore [^18^F]FDG PET/CT could improve the accuracy of the cN staging. cN0 patients assessed with PET and conventional imaging may have a significant risk of occult metastases and a preventive neck dissection of the predicted neck level pathways of the tumor site should be performed [24,25]. A sentinel node biopsy for cN0 patients (cT1 or cT2 stage) is also a good alternative to neck dissection. This technique is less invasive and has demonstrated a similar accuracy for oral cavity cancer [26].

Regarding the sensitivity, [^18^F]FDG PET/CT demonstrated a moderate accuracy ranging from 60 to 63%. These results are in line with earlier findings on the performance of [^18^F]FDG PET/CT for detecting lymph node involvement. A meta-analysis by Sun et al. reported an overall pooled sensitivity of 80% using a similar neck level analysis, but a large heterogeneity in sensitivity was reported for individual studies with values ranging between 31% and 100% [16]. A prospective study including 91 patients with similar characteristics, mostly oral cavity cancer and nodal stage pN0, reported a sensitivity of 69%, similar to the one reported in this study [22]. Regarding subgroup analysis comparing the preoperative cN status, we showed a significantly lower sensitivity of [^18^F]FDG PET/CT in cN0 patients compared to cN+ (40% vs. 75%). This study confirms earlier findings on the lower sensitivity for detecting lymph node involvement in cN0 patients. Indeed, a meta-analysis performed by Kyzas et al. reported a pooled sensitivity of 50% in cN0 patients versus 94% for cN+ [18]. This can be attributed to the fact that patients classified as cN+ have larger metastatic lymph nodes, often palpable and larger than 10 mm compared to patients classified as cN0. The performance of [^18^F]FDG PET/CT is limited by its inherent spatial resolution. Phantom studies performed on a similar Philips Gemini PET/CT system have shown that the recovery coefficient dropped exponentially with smaller sphere diameter, and even stronger for those with a diameter below 10 mm [23]. The latter explains that [^18^F]FDG uptake is strongly underestimated in small metastatic lymph nodes, and hence small lymph node metastases are not detected on PET. Therefore, small or micro metastases can have a low or no significant uptake compared to the jugular internal vein, leading to a visual score of grade 0. This limitation is supported by Figure 4, which illustrates a visual grade 1 uptake in in lymph nodes with a median size of 8 mm and an absent uptake (visual grade 0) for those with a median size of 3 mm. Nevertheless, our data demonstrated that [^18^F]FDG PET/CT detected lymph node metastases in up to 40% of neck levels classified as negative using conventional imaging (i.e., lymph nodes with a short axis < 10mm), and therefore [^18^F]FDG PET/CT could improve the accuracy of the cN staging. cN0 patients assessed with PET and conventional imaging may have a significant risk of occult metastases and a preventive neck dissection of the predicted neck level pathways of the tumor site should be performed [24,25]. A sentinel node biopsy for cN0 patients (cT1 or cT2 stage) is also a good alternative to neck dissection. This technique is less invasive and has demonstrated a similar accuracy for oral cavity cancer [26].

Semi-quantitative [^18^F]FDG PET-derived metrics, SUV max and SUV ratio did not outperform visual analysis for detecting lymph node involvement. Based on these results, visual assessment of cervical lymph node uptake can be encouraged in clinical practice. Moreover, many technical and biological factors (e.g., PET/CT system, reconstruction parameters, incubation time, etc.) can affect standardized uptake values (SUV), which may interfere with the reproducibility and accuracy of quantitative metrics [27].

Some recent studies have demonstrated that PET/MR could improve the accuracy of the workup for evaluating the nodal involvement in HNSCC. By combining the advantages of MRI and [^18^F]FDG PET, the sensitivity and specificity can be increased, detecting, respectively, more metastatic lymph nodes (small and cystic lymph nodes) with fewer false positives (reactive lymph nodes) [28,29,30,31].

Our study suffers from several limitations that are inherently associated with a retrospective study design. Secondly, HNSCC represents a large heterogeneous entity of tumors with disparities regarding biological and clinical behavior as well as their abilities for metastatic spread [32,33]. Thirdly, we performed an analysis comparing [^18^F]FDG uptake in neck levels to the corresponding histological result. This methodology may be prone to some errors in neck level classification between the PET read-out and the surgical neck level localization. Fourthly, we did not analyze the diagnostic accuracy of conventional imaging in these data. Therefore, we cannot demonstrate the superiority of FDG PET/CT compared to enhanced MRI and we cannot highlight the superiority of the combination of both conventional and metabolic imaging for improving the accuracy of the clinical nodal status.

## 5. Conclusions

[^18^F]FDG PET/CT has a high specificity for evaluating the presence of lymph node metastases in HNSCC with similar accuracy using visual or semi-quantitative approaches. Every lymph node with a [^18^F]FDG uptake higher than vascular blood pool localized in the predicted neck pathway of the tumor site must be considered as suspect and cervical lymph node uptake higher than liver background is strongly correlated with malignancy. Therefore, the metabolic activity of lymph nodes must be taken into account during the pre-therapeutic workup and could modify the treatment planning and thus improve the prognostics of HNSCC patients.

## Figures and Tables

**Figure 1 diagnostics-13-02638-f001:**
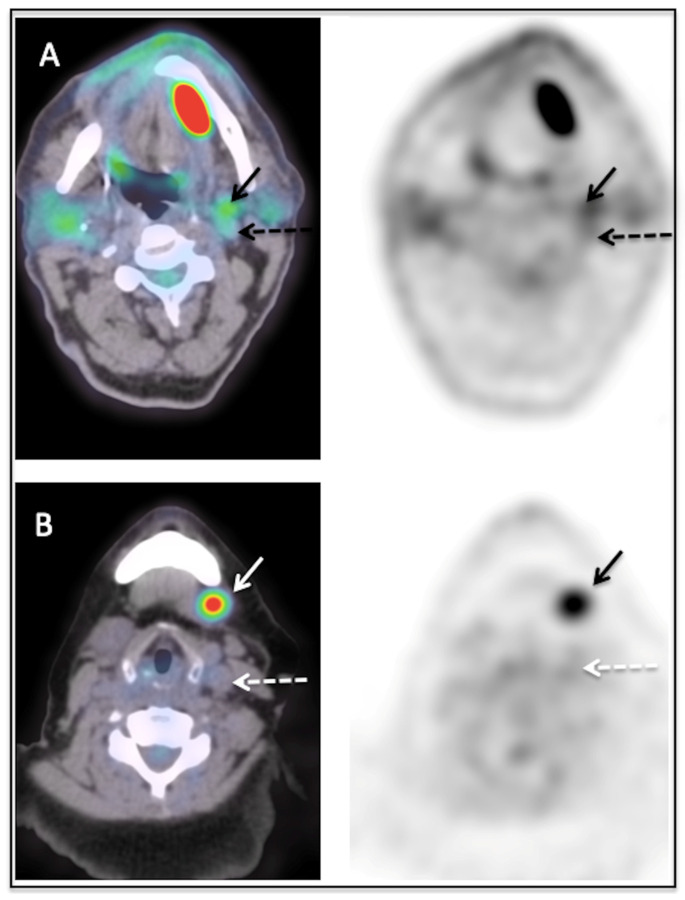
Examples of visual scores 1 (**A**) and 2 (**B**). (**A**) shows a lymph node scored 1 with an uptake (solid arrow) superior to the jugular internal vein (dashed arrow) but inferior to the liver uptake. The SUVmax of this lymph node was 3.8. (**B**) shows a lymph node with a high uptake (solid arrow), superior to the jugular internal vein (dashed arrow) and to the liver uptake. The SUVmax of this lymph node was 7.3.

**Figure 2 diagnostics-13-02638-f002:**
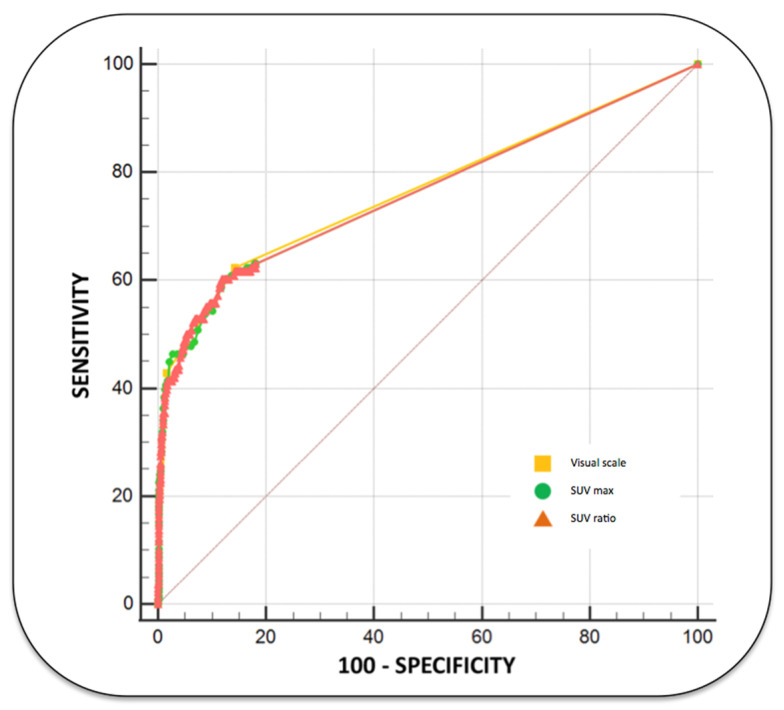
ROC curves demonstrating the global performance of PET for determining neck lymph node metastases using a visual scale, SUV max and SUV ratio. The areas under the ROC curves (AUCs) were similar for each of these parameters (0.762 for SUVmax and SUVR and 0.768 using the visual scale).

**Figure 3 diagnostics-13-02638-f003:**
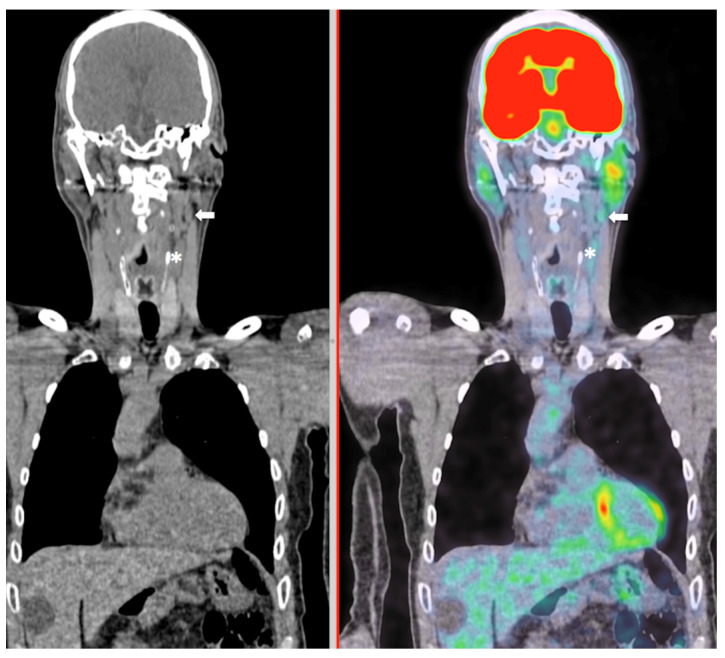
Example of a patient with a left-edge oral cavity cancer and with a [^18^F]FDG uptake in the ispilateral neck level II (white arrow). Note that the [^18^F]FDG uptake was grade 1 (SUVmax = 2.6), slightly superior to the jugular internal vein (white asterisk). The histological report confirmed a metastasis in this neck level.

**Figure 4 diagnostics-13-02638-f004:**
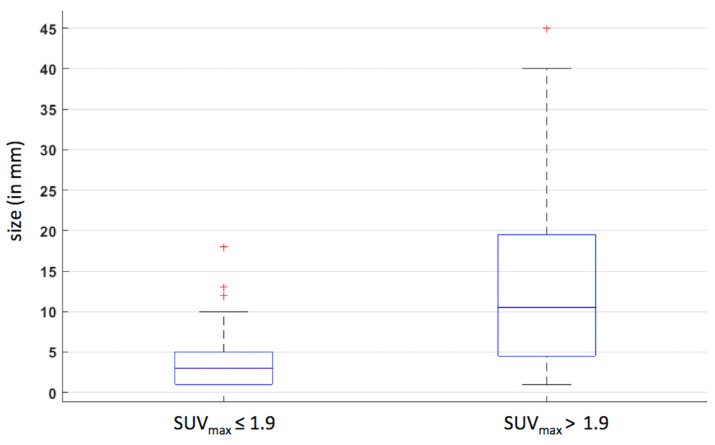
Boxplots evaluating the size of metastases in function of the [^18^F]FDG uptake using a cutoff SUV max: 1.9. Red crosses corresponded to outliers.

**Figure 5 diagnostics-13-02638-f005:**
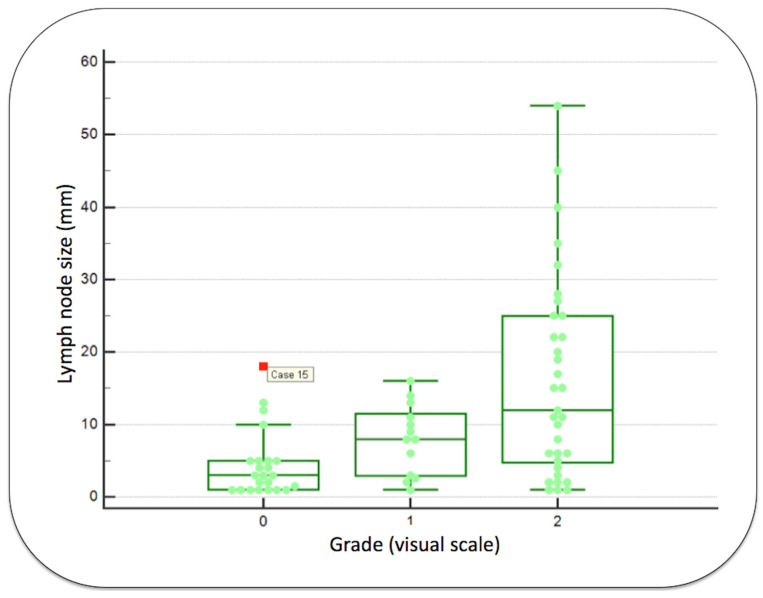
Boxplots evaluating the size of metastases in function of the [^18^F]FDG uptake using a visual scale. Case 15 represents an outlier, a large metastatic lymph node without significant [^18^F]FDG uptake, probably due to the study methodology (errors in reported neck levels after neck dissection).

**Table 1 diagnostics-13-02638-t001:** Patient characteristics.

Site	*n*, %	Male Proportion	Median Age in Years (Range)	Diagnosis (*n*, %)	Recurrence (*n*, %)	cN0 (*n*, %)	cN+ (*n*, %)
Overall	211	72%	61	179	32	144	67
	(100%)		(25–96)	(85%)	(15%)	(68%)	(32%)
Oral cavity	156	74%	60	152	4	102	54
	(74%)		(25–96)	(97%)	(3.0%)	(65%)	(35%)
Larynx	40	73%	63	13	27	32	8
	(19%)		(33–88)	(32%)	(68%)	(80%)	(20%)
Oropharynx	10	50%	65	9	1	8	2
	(5.0%)		(57–74)	(90%)	(10%)	(80%)	(20%)
Others *	5	90%	55	5	0	2	3
	(2.0%)		(40–65)	(100%)	(0.0%)	(40%)	(60%)

* hypopharynx (*n* = 2), nasopharynx (*n* = 1), sinus (*n* = 1), salivary gland (*n* = 1).

**Table 2 diagnostics-13-02638-t002:** Histological characteristics of neck dissections.

	Overall *n* (%)	Oral Cavity *n* (%)	Larynx *n* (%)	Oropharynx *n* (%)	Others * *n* (%)
pT1 (%)	64	51	5	8	0
	(30%)	(33%)	(13%)	(80%)	(0.0%)
pT2 (%)	58	48	5	2	3
	(55%)	(31%)	(13%)	(20%)	(60%)
pT3 (%)	30	27	2	0	1
	(14%)	(17%)	(5.0%)	(0.0%)	(20%)
pT4a (%)	59	30	28	0	1
	(28%)	(19%)	(70%)	(0.0%)	(20%)
pN0 (%)	148	102	36	7	3
	(70%)	(65%)	(90%)	(30%)	(40%)
pN1 or more (%)	63	54	4	3	2
	(30%)	(35%)	(10%)	(30%)	(40%)
Unilateral neck dissection	127	99	14	9	5
	(60%)	(64%)	(35%)	(90%)	(100%)
Bilateral neck dissection	84	57	26	1	0
	(40%)	(36%)	(65%)	(10%)	(0.0%)
Number of dissected levels per patient median [95% CI]	4 [4; 4]	4 [4; 4]	6 [5; 6]	3 [3; 4]	4 [3; 4]
Number of dissected lymph nodes per patient median [95% CI]	32 [29; 35]	32 [29; 35]	33 [28; 46]	22 [11; 39]	47 [21; 58]

* hypopharynx (*n* = 2), nasopharynx (*n* = 1), sinus (*n* = 1), salivary gland (*n* = 1).

**Table 3 diagnostics-13-02638-t003:** Diagnostic performance of [^18^F]FDG PET/CT in the entire cohort.

	Criterion	Se (%) [95% CI]	Sp (%) [95% CI]	AUC [95% CI]	*p*-Value
SUV max	>1.9 *	61	86	0.762 [0.743; 0.780]	<0.0001
		[52; 69]	[85; 88]		
	>3.6 †	38	99		
		[30; 47]	[98; 99]		
SUV ratio	>1.06 *	60	88	0.762 [0.744; 0.780]	<0.0001
		[52; 68]	[87; 90]		
	>2.1 †	31	99		
		[24; 40]	[99; 100]		
Visual scale (grade)	>0 *	63	86	0.768 [0.749; 0.786]	<0.0001
		[54; 71]	[84; 87]		
	>1 †	43	98		
		[34; 52]	[98; 99]		

* Cut-off based on Youden’s J statistic giving equal weight to false positive and false negative values, † cut-off based on maximizing specificity.

**Table 4 diagnostics-13-02638-t004:** SUVmax of lymph nodes classified as benign or malignant based on histology.

	Liver	Benign Lymph Nodes * (*n* = 839)	Malignant Lymph Nodes (*n* = 138)	*p*-Value
SUVmax Mean (range)	2.9 (1.7–4.5)	2 (0.9–12.7)	3.6 (1.2–23.7)	<0.001

* Lymph nodes without proven metastases at histology.

**Table 5 diagnostics-13-02638-t005:** Diagnostic performance of [^18^F]FDG PET/CT in patients with cN0 and cN+.

	Criterion	Se (%) [95% CI]	Sp (%) [95% CI]	AUC [95% CI]	*p*-Value
cN0 (*n* = 144)					
SUV max	>2.0 *	38	89	0.641	0.0007
		[24; 54]	[88; 91]	[0.616; 0.666]	
SUV ratio	>1.08 *	38	89	0.638	0.0007
		[24; 54]	[88; 91]	[0.612; 0.663]	
Visual scale (grade)	>0 *	40	85	0.637	0.0006
		[26; 56]	[83; 87]	[0.611; 0.662]	
cN+ (*n* = 67)					
SUV max	>1.5 *	73	84	0.814	<0.0001
		[63; 82]	[80; 87]	[0.782; 0.843]	
SUV ratio	>0.92 *	73	85	0.816	<0.0001
		[63; 82]	[81; 88]	[0.785; 0.845]	
Visual scale (grade)	>0 *	74	83	0.816	<0.0001
		[64; 83]	[80; 86]	[0.784; 0.844]	

* Cut-off based on Youden’s J statistic giving equal weight to false positive and false negative values.

## Data Availability

The data presented in this study are available on request from the corresponding author.

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
