# Peer review of "Comparable Accuracy of Quantitative and Visual Analyses of [18F]FDG PET/CT for the Detection of Lymph Node Metastases from Head and Neck Squamous Cell Carcinoma"

_diagnostics, 2023, doi:10.3390/diagnostics13162638_

Round 1

Reviewer 1 Report

The role of PET/CT scan in detecting neck metastases, recurrent tumour and distant metastases of HNSCC is well known.

The manuscript discuss on the neck metastases only and the data presented is not robust.

Several revisions are necessary as in the yellow highlight of commented pdf text.

Thank you.

The role of PET/CT scan in detecting neck metastases, recurrent tumour and distant metastases of HNSCC is well known.

The manuscript discuss on the neck metastases only and the data presented is not robust.

Several revisions are necessary as in the yellow highlight of commented pdf text.

Thank you.

Author Response

Dear reviewer,

We thank you for your consideration and for your interesting comments. It is a point to point report.

  1. “Please revise the title “with high specificity lymph node “.(line 2-3)”

Answer: The title was changed as follows: [18F]FDG PET/CT detects accurately lymph node metastases from head and neck squamous cell carcinoma.

  1. This statement is inaccurate, as PETCT can also be used to detect local recurrent disease, especially in post chemoradiation cases where the fibrosis can be intermindgle with tumor.(line 17-18)

Answer:  We added this comment in the manuscript as follows:

“[18F]FDG PET/CT is recommended for detecting recurrent disease and in the initial staging for evaluating distant metastases but its interest for detecting cervical lymph metastases remains unclear.”

  1. “similar performances ” is unclear, please revise.(line 26)

Answer: It is changed by “diagnostic accuracy”.

  1. “a significant comorbidity” is inaccurate, please revise.(line 60-61)

Answer: The sentence was changed as follows (and with new references):

“Selective neck dissection maximizes disease free survival but is associated with a significant risk of complications and possible impaired quality of life (14, 15).”

  1. Why spacing?(line63-69)

Answer: Modified.

  1. This visual analog assessment is unclear. Please add examples of cases.

Also do include in the discussion segment whether this method is sensitive or not, if so, why?(line89-95)

Answer: A new figure was added demonstrating examples of visual score grade 1 and 2 (new figure 1). A more precise definition of this visual score was also added a follows (lines 100-106):

 “First, a visual analysis was performed using a 3-point grading scale, comparing the lymph node uptake to those of the internal jugular vein and of the liver. A lymph node classified grade 0 had no significant uptake (≤ blood pool activity in internal jugular vein), a lymph node classified grade 1 had a low uptake (> blood pool activity in internal jugular vein and ≤ liver uptake) and a lymph node classified grade 2 had a high uptake (> liver uptake). Figure 1 illustrates examples of lymph nodes classified grade 1 (Figure 1A) and grade 2 (Figure 1B).”

  1. These data is not robust enough. Please add the exact SUV value of nodes for all cases in the studied cohort.(line147-152)

Answer: Indeed, it is a very interesting remark. We added a new table (table 4) and this text (lines 185-189):

“Regarding the whole cohort, the SUVmax of lymph nodes with proven metastases at histology were significantly higher than the SUVmax of lymph nodes without proven metastases (Table 4). Note that the SUVmax of malignant lymph nodes were generally higher than the SUVmax of the liver and the SUVmax of benign lymph nodes generally lower than the SUVmax of the liver (comparing the means). “

  1. Please specify the exact SUV value of this node.(line157,158)

Answer: “Added in the manuscript “SUVmax= 2.6”

9.How do you confirm it is malignancy if SUV is higher, as inflammation of nodes also can have high SUV.

How to exclude False positive results? Please discuss.(line192-195)

Answer: Thank you for this remark, it is indeed one of the interests of this study to find a way to discreminate reactive to malignant lymph nodes. The first part of the discussion was modified as follows:

“In a large cohort, this study demonstrated the high specificity of [18F]FDG PET/CT for detecting lymph node metastases with a specificity of 86% using a visual score grade 1 and a specificity of 98% for a grade 2. These results are concordant with previous studies reporting specificities ranging from 86% to 96% (16, 18). As such, our study shows that every lymph node with an uptake higher than vascular blood pool in a predicted pathway neck level of the primary tumor must be considered as suspect and that a visual grade 2 is strongly correlated with malignancy. In our data, the false positive rate was only 1-2% for lymph nodes with high [18F]FDG uptake (SUVmax > 3.6 or visual score grade 2). For lymph nodes with low [18F]FDG uptake (SUVmax > 1.9 or visual score grade 1), the false positive rate increased to 14%. Therefore, the higher the [18F]FDG uptake in lymph nodes is and the higher is the probability of the malignancy (more the malignancy is highly probable) . In lymph nodes with low [18F]FDG uptake, false positive results due to reactive lymph nodes cannot be excluded. Our results are concordant with the study of Nakagawa et al., demonstrating a low [18F]FDG uptake in reactive lymph nodes (mean SUVmax= 2.2)(21). “

  1. Please discuss in detail the VAS assessment whether it sensitive or not? And if so, why?(line216-218)

Answer: The problems linked to the moderate sensitivity were discussed in the second paragraph of the discussion (lines 235-266). The VAS has a moderate sensitivity equal to SUVmax and SUV ratio. It is mainly due to micrometastases undetectable with [18F]FDG PET/CT. For improving the understanding, this sentence was added in the manuscript (lines 255-256):

“Therefore, small or micro metastases can have a low or no significant uptake compared to the jugular internal vein, leading to a visual score grade 0.”

  1. Please revise the conclusion to include the significance of positive PET/CT for neck metastases and its impact on subsequent HNSCC treatment.

Answer: We added this sentence (line291-293):

“Therefore, the metabolic activity of lymph nodes must be taken into account during the pre-therapeutic workup and could modify the treatment planning and thus improve the prognostic of HNSCC patients. “

  1. The literature is suboptimal. Please add more recent literatures and cites within the text accordingly.(line268-327).

Answer: More recent references were added in the manuscript;  references 10, 11, 14, 15 and 21.

Reviewer 2 Report

This is an interesting study about the role of PET/CT for lymph node metastases from head and neck squamous cell carcinoma. The authors found a specificity of 83-98% and a sensitivity of 60-63%.

The paper is well written. However, some issues remain.

Please report which conventional imaging was used for staging.

cN+ based on conventional imaging was defined on nodal short axes. Did the authors use any other criteria (e.g., nodal colliquation, etc.)?

More detailed clinical and pathological nodal staging should be reported and used in the analyses.

The p16 status must be reported. It may be interesting to introduce such variable in the analyses.

Please report sensibility and specificity of conventional imaging both in the Introduction (data from literature) and the Results (data from this case series). Moreover, compare such information with data from PET/CT. This information is necessary to understand the role of PET/TC for node metastases and achieve correct conclusions.

Author Response

Dear reviewer,

We thank you very much for your review and your interesting comments.

This is point by point response to your comments:

  • “Please report which conventional imaging was used for staging.

cN+ based on conventional imaging was defined on nodal short axes. Did the authors use any other criteria (e.g., nodal colliquation, etc.)?”

Response highlighted in green in the manuscript (end of page 2):

"Gadolinium-enhanced MRI of the neck was performed for each patient. Patients classified cN+ had at least one lymph node with a short axis upper than 10 mm. No other criterion was used for determining the cN status."

  • “More detailed clinical and pathological nodal staging should be reported and used in the analyses. The p16 status must be reported. It may be interesting to introduce such variable in the analyses.”

Response: This retrospective study analyzed patients from 2007 to 2019. For many patients, especially for patients treated before 2010, no information regarding the P16 status was found in the medical record and therefore we cannot do this type of report. We tried to report accurately the clinical data regarding the cN status for each tumor type (table 1). We tried as well to show an extensive report of the neck dissections, comprising the pN status for each tumor type (table 2).

  • “Please report sensibility and specificity of conventional imaging both in the Introduction (data from literature) and the Results (data from this case series). Moreover, compare such information with data from PET/CT. This information is necessary to understand the role of PET/CT for node metastases and achieve correct conclusions.”

Response highlighted in green in the manuscript (page 2):

"The recent literature reports a variable diagnostic accuracy of conventional imaging with a sensitivity and specicificity ranging from 77% to 100% using enhanced MRI (11). Despite significant improvements in conventional imaging, the overall rate of occult micrometastases, undetectable by conventional imaging, ranges between 10% to 55% (12).This text was added in the manuscrpit in the introduction (P2, highlighted in green).

Regarding our data, the percentage of patients classified cN postive based on conventional imaging was 32% (table 1) and the percentage of patients classified pN positive based on histological results was 30% (table 2). Many variations can occur in the different neck stations comparing PET/CT to conventional imagaing (a small lymph node can have a strong FDG uptake and a big reactive lymph node can have a low FDG uptake. This analyse was not perfomed because it was not the aim the study. We analysed more than 800 neck levels in this data, it is impossible to perform a second analysis of the diagnostic performances of conventional imaging for each of the neck levels. Nevertheless, it is indeed a limitation of the study and we add this text at the end of manuscript (highlighted in green, page 10) :

« Fourthly, we did not analyze the diagnostic accuracy of conventional imaging in this data. Therefore we cannot demonstrate a superiority of FDG PET/CT compared to enhanced MRI and we cannot highlight the superiority of the combination of both conventional and metabolic imaging for improving the accuracy of the clinical nodal status. »

Round 2

Reviewer 2 Report

Thanks for improving the manuscript.

Author Response

Thank you very much for your time and your input